# Changes in Interleukin-1β, Tumor Necrosis Factor-α, and Interleukin-10 Cytokines in Older People with Periodontitis

**DOI:** 10.3390/geriatrics8040079

**Published:** 2023-08-10

**Authors:** Ines Augustina Sumbayak, Sri Lelyati C. Masulili, Fatimah Maria Tadjoedin, Benso Sulijaya, Arrum Mutiara, Diana Khoirowati, Yuniarti Soeroso, Boy M. Bachtiar

**Affiliations:** 1Department of Periodontics, Faculty of Dentistry, Universitas Indonesia, Jakarta 10430, Indonesia; inessumbayak@gmail.com (I.A.S.); fatimah.tadjoedin@ui.ac.id (F.M.T.); bensosulijaya@gmail.com (B.S.); tiashergi@gmail.com (A.M.); dianachoiro@yahoo.com (D.K.); yuniarti_22@yahoo.co.id (Y.S.); 2Department of Oral Biology, Faculty of Dentistry, Universitas Indonesia, Jakarta 10430, Indonesia; boy_mb@ui.ac.id

**Keywords:** periodontal disease, periodontitis, aging, inflammation, cytokine

## Abstract

Aging can change the ability to respond to various stimuli and physical conditions. A decreased immune response is a form of deterioration of function in older people, who then become more vulnerable when exposed to pathogens. Periodontitis is an inflammatory disease of the periodontal tissues that often occurs in older people. This study aimed to clinically analyze the periodontal status and cytokine levels of IL-1β, TNF-α, and IL-10 in older people and adults with periodontitis. This clinical study examined 20 persons in a group of older people and 20 persons in a group of adults. The clinical measurements of periodontal status included the Simplified Oral Hygiene Index (OHI-S), Plaque Index (PlI), and Papilla Bleeding Index (PBI). The cytokine levels in gingival crevicular fluid (GCF) were quantified by using ELISA kits. The OHI-S, PlI, and PBI were found to be higher in the older group. The mean values of cytokines were higher in the older group than in adults, although no statistically significant differences were found. A strong correlation was found between the clinical measurements and the cytokine levels in the GCF. There was an increasing tendency of pro-inflammatory and anti-inflammatory cytokines in the older group compared to the adult group.

## 1. Introduction

Periodontitis is an inflammatory disease of periodontal tissues, leading to the progressive loss of bone support for the teeth [1,2]. It is caused by the accumulation of dysbiotic biofilms on the surface of the teeth [1,3,4]. Dysbiotic biofilms are formed when the normal balance of bacteria in the oral cavity becomes disrupted, leading to an overgrowth of harmful bacteria and a decrease in beneficial bacteria [3,4,5]. The bacteria in dysbiotic biofilms release toxins and by-products that irritate and damage the periodontal tissues [4,5,6]. The blood vessels in the affected area dilate (vasodilation), leading to increased blood flow to the site of infection [1,4,6]. This increased blood flow allows immune cells and antibodies to reach the infected area quickly [1,7]. The immune cells release various inflammatory mediators, such as cytokines and prostaglandins [1,7,8].

Cytokines are a diverse group of small proteins that are produced by a wide range of cells, including immune cells such as T cells, B cells, macrophages, and dendritic cells, as well as non-immune cells like fibroblasts and endothelial cells [7,8,9]. The main function of cytokines is to regulate and modulate the immune response by facilitating communication between different cells [1,7,8]. They can act as both pro-inflammatory and anti-inflammatory agents [1,9]. In the periodontium, several cytokines play essential roles in the immune response and inflammation [7,8]. Three of the key cytokines involved are cytokines interleukin-1 beta (IL-1β), tumor necrosis factor-alpha (TNF-α), and interleukin-10 (IL-10) [10,11,12]. IL-1β is a critical mediator of the inflammatory response [1,10]. It promotes the recruitment and activation of immune cells, such as neutrophils, and contributes to tissue destruction by inducing the release of enzymes that degrade the extracellular matrix in the periodontium [10,13]. TNF-α plays a crucial role to induce the expression of other pro-inflammatory molecules, activates immune cells, and promotes the destruction of periodontal tissues [7,10]. IL-10, in contrast to IL-1β and TNF-α, is an anti-inflammatory cytokine. It inhibits the production of pro-inflammatory cytokines and reduces the activation of immune cells that contribute to tissue destruction [12,14].

The balance between pro-inflammatory cytokines like IL-1β and TNF-α and anti-inflammatory cytokines like IL-10 is crucial for maintaining periodontal health [15]. Dysregulation of this balance can lead to chronic inflammation and tissue destruction, which are characteristic features of periodontal diseases such as gingivitis and periodontitis [1,15]. The studies conducted by Passoja et al. and Acharya et al. showed that periodontitis patients had a significantly lower level of IL-10 than the healthy group [14,16]. On the other hand, a study by Toker et al. demonstrated a contradictory result, which showed that increasing the level of pro-inflammatory cytokines in periodontitis patients promoted anti-inflammatory escalation [12].

Aging is associated with changes in the immune system, termed immunosenescence [17]. Immunosenescence refers to the gradual deterioration of the immune system that occurs with aging, leading to an increased susceptibility to infections, a reduced ability to generate protective immune responses to vaccines, and an increased risk of certain diseases [17,18,19]. Hazeldine et al. and Montgomory et al. found that cytokine levels in older people were significantly reduced due to the decline in immune function [20,21]. However, Franceschi et al. and Clark et al. demonstrated that the immune system in older people produces more pro-inflammatory cytokines [22,23]. Inflammaging is a term used to describe the chronic low-grade inflammation that occurs as a part of the aging process [17,22,24]. The concept suggests that, as people age, their bodies experience a gradual increase in pro-inflammatory factors, leading to a state of persistent, low-level inflammation [22,24]. This phenomenon leads to alterations in the production and regulation of cytokines in response to microbial challenges in the oral cavity [22,23,24]. As a result, the immune system may become less effective in controlling inflammatory processes in the periodontium, leading to a higher susceptibility to periodontal diseases [17,22].

It is important to note that, while there may be age-related changes in cytokine levels, periodontitis is a multifactorial disease [1,3]. Other factors, such as oral hygiene practices and overall health status, also play significant roles in the development and progression of periodontitis [4,25]. Therefore, we hypothesized that there might be an association between age-related changes in cytokine levels and periodontal status in periodontitis patients. The periodontal status included the clinical measurements of Simplified Oral Hygiene Index (OHI-S), Plaque Index (PlI), and Papilla Bleeding Index (PBI). In this study, we aimed to clinically analyze the periodontal status and the cytokine levels of IL-1β, TNF-α, and IL-10 in older people and adults with periodontitis.

## 2. Materials and Methods

### 2.1. Ethical Clearance

This cross-sectional study was approved by the Ethical Committee of Dental Research, Faculty of Dentistry, Universitas Indonesia (No. 30/Ethical Approval/FKGUI/IX/2020).

### 2.2. Subjects

This study included 40 subjects divided into two groups: a 20-person older group (60–74 years old) and a 20-person adult group (35–45 years old). The formula used to calculate the sample size was based on previous research by Tsalikis to determine the value of the difference in means between group 1 and group 2 and the combined standard deviation [26]. We found the minimum sample size for this study was 19 persons in each group using this formula, and we rounded up the sample size to 20 persons each group. The inclusion criteria were subjects diagnosed with Stage III Grade B Periodontitis based on AAP/EFP 2017 [2]. Periodontitis was defined based on clinical attachment loss (CAL) of ≥2 mm of ≥2 non-adjacent teeth, bleeding on probing (BoP), and presented a probing pocket depth (PPD) of 5–7 mm as measured using UNC-15 periodontal probe (Hu-Friedy, Chicago, IL, USA) [2]. The exclusion criteria were subjects diagnosed with diabetes, motoric and cognitive disorders, human immunodeficiency virus infection, pregnancy, those smoking in any form, those with current orthodontic treatments, and those with a history of prophylactic antibiotic use in the last 3 months. Written informed consent was provided to all subjects.

### 2.3. Examination of Clinical Measurements

The examination of periodontal status was performed by two examiners (IAS and DK) who assessed the following clinical measurements:-OHI-S index system developed by Greene and Vermillion (1960). This consists of two main components: debris index (DI) and calculus index (CI). The scores from both indices are added to calculate the OHI-S score for each individual. The overall OHI-S score can range between 0 and 6. Scores of 0–1.2 = good oral hygiene. Scores of 1.3–3 = moderate oral hygiene. Scores of 3.1–6 = poor oral hygiene [27,28].-PlI index system developed by Löe and Silness (1964). This evaluates the thickness of dysbiotic biofilms at the gingival margin on certain tooth surfaces. Score 0 = no dysbiotic biofilms present. Score 1 = a thin layer of dysbiotic biofilms that can be seen by running a probe across the tooth surface. Score 2 = moderate accumulation of dysbiotic biofilms. Score 3 = abundance of dysbiotic biofilms [25,27].-PBI index system developed by Muhlemann (1977). This evaluates the bleeding tendency of the gingival tissues when gently probed. The index is scored based on the presence or absence of bleeding. Score 1 = a single bleeding point appears. Score 2 = a fine line of blood or several bleeding points appears. Score 3 = interdental triangle becomes more filled with blood. Score 4 = immediately after probing, blood flows into the interdental area to cover portions of the tooth and/or gingiva [27,29].

### 2.4. GCF Sample Collection

GCF was collected in each patient on teeth with a probing depth of 5–7 mm [30]. The sites were isolated with cotton rolls, and gently dried with a moisture-free stream of air. Paper points #30 (Gapadent, Tianjin, China) were inserted gently into the gingival crevice for 30 s [30,31]. This procedure was carried out three times. Paper points visibly contaminated with blood were discarded [30,31]. The three paper points were placed in Eppendorf tubes containing 1000 μL of phosphate-buffered saline (Gibco, Waltham, MA, USA) [31]. The samples were then centrifuged and stored at −20 °C in the Integrated Laboratory, Faculty of Medicine, Universitas Indonesia for further processing [30].

### 2.5. IL-1β, TNF-α, and IL-10 Assays

ELISA was used to measure the cytokine levels [32]. All samples and standards were assayed in duplicate [32]. The IL-1β, TNF-α, and IL-10 levels were quantified by using Milliplex^®®^ Map Human Cytokines ELISA kits (Merck, Gillingham, UK). Twenty samples were analyzed for each group (total of 40 samples) for the determination of the cytokine levels.

### 2.6. Statistical Analysis

Data analyses for both clinical and in vitro studies were performed with SPSS Statistics Version 23 (IBM Corporation, New York, NY, USA). The mean value of each variable in the clinical study was calculated. A Mann–Whitney test was used for the comparative analysis of cytokine levels and clinical measurements between age categories. Spearman’s test was also performed to calculate association between clinical measurements and cytokine levels in GCF. Statistical significance was set at *p* < 0.05.

## 3. Results

### 3.1. Data Distribution

Demographic data with the minimum–maximum and median values of normality test with *p* > 0.05 for data distribution are presented in Table 1. The percentage of females was higher than that of males in the older group, while the percentage of females was the same as that of males in the adult group.

The distribution of the clinical measurements and cytokine levels in the older and adult groups are presented in Table 2. Both groups showed a median value of the OHI-S greater than 3.00 (poor) and PlI greater than 2.1 (poor). Based on the minimum and maximum values of the PBI, both groups were classified into the category of moderate to severe inflammation. The median value of the cytokine levels of IL-1β, TNF-α, and IL-10 in the older group was higher than that in the adult group.

### 3.2. Comparison of Clinical Measurements and Cytokine Levels

A comparative analysis of the clinical measurements showed statistically significant differences (*p* < 0.05) between the age categories (Figure 1). The elderly group had higher values of OHIS, PlI, and PBI than the adult group. Figure 2 presents a comparative analysis of cytokine levels between the age categories. The mean value of the cytokine levels of IL-1β, TNF-α, and IL-10 in the older group was higher than that in the adult group, although no statistically significant differences were found (*p* > 0.05).

### 3.3. Correlation between Clinical Measurements and Cytokine Levels

Table 3 and Table 4 present the correlation between clinical measurements and cytokine levels in age categories. Both groups showed a strong, positive, and statistically significant linear correlation between OHIS, PlI, PBI and the cytokine levels of IL-1β, TNF-α, and IL-10 (r > 0.5 and *p* < 0.05). The high values of OHIS, PlI, and PBI corresponded to the increased cytokine levels of IL-1β, TNF-α, and IL-10.

## 4. Discussion

Forty subjects were divided into two groups based on age categories as defined by the WHO: a 20-person older group (60–74 years old) and a 20-person adult group (35–45 years old) [33]. Subjects were selected who were diagnosed with Stage III Grade B Periodontitis based on AAP/EFP 2017 [2]. Tadjoedin et al. showed that periodontitis was found in as many as 56% in the adult group, increasing to 88% in the older group [34].

Subjects diagnosed with diabetes, human immunodeficiency virus infection, pregnancy, those smoking in any form, those with current orthodontic treatments, and those with a history of prophylactic antibiotic use in the last 3 months were considered as exclusion criteria because those conditions were proven to affect the changes of immune response to pathogens [35]. Older people who had lost all their teeth and those with motor and cognitive disorders were not included because their experiences in oral hygiene were not in the same condition as older people who still had their teeth and those without motor and cognitive disorder [36,37].

Table 1 presented the distribution of subjects based on age and sex. The age of the older group had a median value of 67 years old, with 12 female subjects and 8 male subjects. The adult group had a median value of 39 years old, with 10 female subjects and 10 male subjects. The older group had more women than men, while the adult group had an equal number of women and men. Hormonal changes throughout a woman’s life, such as during puberty, menstruation, pregnancy, and menopause, can affect the oral environment and gingival health [38,39]. During menopause, decreased estrogen levels may lead to increased inflammation and bone loss in the periodontium [39]. However, we did not assess the relationship between gender and inflammaging in our study.

Measurement of periodontal status in this study included clinical measurements of OHI-S, PlI, and PBI. OHI-S, developed by John C. Greene and James R. Vermillion, is a widely used method used to assess and measure oral hygiene in individuals [28]. It is used to determine the presence of debris and calculus on tooth surfaces, providing a quantitative measure of oral hygiene [27,28]. PI, developed by Löe and Silness, is a numerical score developed by Löe and Silness to assess the amount of dysbiotic biofilms on tooth surfaces [27]. The higher the PlI score, the more dysbiotic biofilms are present, indicating a higher risk of periodontal disease development [25]. PBI, developed by Muhlemann, is used to assess periodontal tissue inflammation and involves examining the bleeding tendency of interdental papilla. The presence or absence of bleeding indicates the level of inflammation in that area [27,29].

Figure 1 shows that the mean values of OHIS, PlI, and PBI were higher in the older group than the adult group. This study result proved that aging has impacts on periodontal status [23,24,40]. As people get older, they may experience changes in their oral health, which can be influenced by various factors, including age-related physiological changes, systemic health conditions, and lifestyle habits [24,40,41]. Older people become more susceptible to periodontal disease. This increased susceptibility is partly due to the cumulative effects of dysbiotic biofilms and oral bacteria over time [3,19,24]. Maintaining good oral hygiene becomes more challenging for some older people. The declining motor skills of older people can make brushing and flossing teeth difficult, leading to inadequate dysbiotic biofilms removal [3].

The clinical symptoms of periodontal disease show the signs of inflammation in periodontal tissue including hiperemia, edema, and bleeding on the gingiva [1]. The presence of interdental papilla bleeding was examined using PBI. Figure 1 showed that PBI in the older group was higher than in adult group. This proves that aging can have a significant impact on inflammation in periodontal tissues [19,22]. Aging can lead to a gradual decline in immune function, which impairs the body’s ability to control and resolve inflammation effectively [23,40]. As a result, the immune response to pathogens in the oral cavity may be compromised, allowing periodontal disease to progress more easily [3,42]. Inflammation is a critical part of the immune response to infection and injury. Aging can alter the function of immune cells, such as neutrophils and macrophages, which play a vital role in the inflammatory response [17,43]. This phenomenon leads to alterations in the cytokine levels in response to microbial challenges in the periodontal tissue [3,18,21].

Furthermore, Figure 2 shows that there were no statistically significant differences in cytokine levels between the older and adult group, although the mean values of cytokine levels in the older group showed a tendency to increase compared with those in the adult group. This study result is in line with studies by Franceschi et al. and Clark et al., who reported an increase in cytokine levels in the older people [22,23]. There are complex changes that occur in the immune system during aging. As a person ages, the effectiveness of the innate immune response tends to decrease [18,22]. Despite the dampening of innate immune responses, there are some signaling pathways within the immune system that show a paradoxical increase in activity as a person ages; specific cytokines actually show an increase in levels as people age [21,43]. This increase in certain cytokines might contribute to age-related inflammatory conditions or diseases [19,44]. Although both the older and adult groups had the same condition of periodontitis, aging could affect the immune response by increasing inflammation. As a result, the immune system may become less effective in controlling inflammatory processes in the periodontium, leading to a higher susceptibility to periodontal diseases [17,23,24].

Aging can be characterized by changes in the immune system in terms of quantity and quality, termed immunosenescence [17,43]. Immunosenescence is a term used to describe the gradual decline of the immune system’s function as a natural consequence of aging [17,21,43]. The word is a combination of “immune” (referring to the immune system) and “senescence” (meaning the process of aging). As people age, their immune system undergoes changes that can result in reduced efficiency and responsiveness. The concept of the “network theory of aging” presented by Franceschi states that a decrease in the ability of a host’s response system to cope with various stresses and antigenic stimuli usually occurs simultaneously with the process of increasing inflammation, which is called inflammaging [22,44].

Inflammaging is a term used to describe the chronic low-grade inflammation that occurs as a part of the aging process [19,24,44]. It combines two words: “inflammation” and “aging” [40]. The exact mechanisms behind inflammaging are not fully understood, but it is thought to be a complex interplay of various factors. Additionally, the immune system’s functionality tends to decline with age, leading to a less-efficient control of inflammatory responses [21,22,44]. This is also considered to have a long cumulative effect of exposure to bacteria present in periodontal tissue [3,19,24]. A decrease in immune responses—both innate and adaptive immunity—along with chronic inflammation, results in changes in immune capabilities and an increase in the number of pathogens in periodontal tissue [17,20,43].

Cytokines have important roles in the inflammatory response; they are synthesized by various types of cell, such as neutrophils, macrophages, lymphocytes, fibroblasts, and epithelial cells [1,8]. The cytokine IL-1β is considered a key pro-inflammatory mediator and is related to the innate immune system. It has the function of stimulating and secreting other inflammatory mediators that contribute to inflammatory processes and tissue destruction [10,13,31]. The cytokine TNF-α is another pro-inflammatory mediator in periodontal disease that is produced primarily by activated macrophages in response to infection [7,10]. TNF-α plays a role in increasing neutrophil activity, mediates cell and tissue changes by stimulating matrix metalloproteinase (MMP) secretion, and increases the production of collagenase and prostaglandin enzymes (PGE2), thereby triggering collagen damage and bone resorption [1,7]. Increased levels of IL-1β and TNF-α in the GCF occur in periodontitis [7,8,9].

Cytokine IL-10 is an anti-inflammatory cytokine that functions to reduce inflammatory responses and can be found clinically in the GCF of patients with periodontal disease [12,14]. IL-10 is thought to have an important role in limiting the duration and extent of inflammatory responses. It is also involved in reducing the process of tissue destruction and decreasing MMP production by increasing tissue synthesis, which inhibits MMP production by macrophages [1,12]. Imbalance of pro-inflammatory and anti-inflammatory cytokines can lead to chronic inflammation and periodontal tissue destruction [1,15]. Along with a study by Toker et al., this study result shows the increasing level of pro-inflammatory cytokines in periodontitis patients promoted anti-inflammatory escalation [12].

The correlation between the clinical measurements and cytokine levels in the GCF was statistically significant (Table 3 and Table 4). The increase in cytokine levels was also related to the clinical examinations. In this study, a higher value of clinical measurements coincided with an increase in IL-1β, TNF-α, and IL-10 levels in the GCF. It proved that the cytokine levels are associated with the periodontal status of patients with periodontitis. This study is in line with a study conducted by Fujita et al., which found that there is a strong correlation between levels of cytokines such as PlI and BoP and periodontal status in patients with periodontitis [45]. Poor oral hygiene will affect the increase in inflammatory mediators, leading to signs of inflammation in periodontal tissue including hiperemia, edema, and bleeding on the gingiva [7,25].

This study’s limitations encourage further study with a greater number of subjects and microbiome analyses performed on the GCF to match the cytokine data with the clinical and microbiome data.

## 5. Conclusions

Aging affects the periodontal status and inflammatory response in patients with periodontitis. The results of this study demonstrate cytokine dynamics of IL-1β, TNF-α, and IL-10 in periodontitis, especially in older people. The strong positive correlation between periodontal status and cytokine levels suggests the adjunctive periodontal therapy for older people.

## Figures and Tables

**Figure 1 geriatrics-08-00079-f001:**
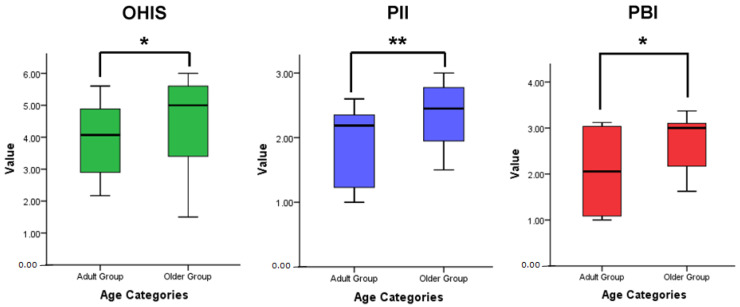
Comparison of the clinical measurements between the older and adult groups with a Mann–Whitney test; * *p* < 0.05 significance, ** *p* < 0.01 significance, *n* = 20 older subjects and 20 adult subjects.

**Figure 2 geriatrics-08-00079-f002:**
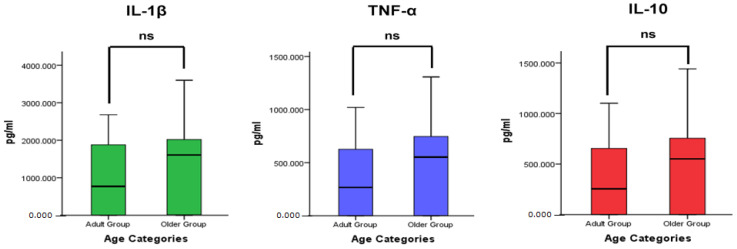
Comparison of the cytokine levels in the GCF of older and adult group with a Mann–Whitney test; *p* < 0.05 significance, ns = not significant (*p* > 0.05), *n* = 20 older subjects and 20 adult subjects.

**Table 1 geriatrics-08-00079-t001:** Distribution of research subjects by age.

Research Subject (*n* = 40)	Total (*n*)	Percentage	Min–Max	Median
Older Group (60–74 years old)			60–73	67
Female	12	60%		
Male	8	40%		
Adult Group (35–45 years old)			35–45	39
Female	10	50%		
Male	10	50%		

Normality test: Shapiro–Wilk.

**Table 2 geriatrics-08-00079-t002:** Distribution of the clinical measurements and cytokine levels in the older and adult groups.

Periodontitis	Age Group	Min–Max	Median
Periodontal Status			
OHI-S	Older Group	1.50–6.00	5.00
	Adult Group	2.17–5.60	4.07
PlI	Older Group	1.50–3.00	2.45
	Adult Group	1.00–2.60	2.18
PBI	Older Group	1.63–3.37	3.00
	Adult Group	1.00–3.12	2.05
Cytokine Levels (pg/mL)			
IL-1β	Older Group	1.46–3597.87	1607.80
	Adult Group	0.28–2679.11	770.02
TNF-α	Older Group	0.24–1306.62	551.97
	Adult Group	0.24–1020.05	266.86
IL-10	Older Group	0.15–1441.36	550.94
	Adult Group	0.15–1101.26	255.07

Normality test: Shapiro–Wilk.

**Table 3 geriatrics-08-00079-t003:** Correlation between clinical measurements and cytokine levels in the older group.

Cytokine	Clinical Measurement					
	OHI-S		PlI		PBI	
	r	*p*	r	*p*	r	*p*
IL-1β	0.650	0.002 *	0.566	0.009 *	0.753	0.00 *
TNF-α	0.685	0.001 *	0.599	0.005 *	0.748	0.00 *
IL-10	0.669	0.001 *	0.579	0.007 *	0.748	0.00 *

Spearman’s test; r = correlation coefficient, * *p* < 0.05 significance.

**Table 4 geriatrics-08-00079-t004:** Correlation between clinical measurements and cytokine levels in the adult group.

Cytokine	Clinical Measurement					
	OHIS		PlI		PBI	
	r	*p*	r	*p*	r	*p*
IL-1β	0.764	0.00 *	0.763	0.00 *	0.799	0.00 *
TNF-α	0.783	0.00 *	0.771	0.00 *	0.834	0.00 *
IL-10	0.784	0.00 *	0.815	0.00 *	0.849	0.00 *

Spearman’s test; r = correlation coefficient, * *p* < 0.05 significance.

## Data Availability

All data will be made available upon reasonable request.

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
