# Peer review of "Changes in Interleukin-1β, Tumor Necrosis Factor-α, and Interleukin-10 Cytokines in Older People with Periodontitis"

_geriatrics, 2023, doi:10.3390/geriatrics8040079_

Round 1
Reviewer 1 Report
Brief summary
This study aimed to analyze the periodontal status and cytokine levels of IL-1β, TNF-α, and IL-10 in an aging model, focusing on the vulnerability of older individuals to periodontitis. The study involved 20 adults and 20 older individuals with periodontitis, examining clinical measurements such as oral hygiene index, plaque index, and papilla bleeding index, as well as cytokine levels in gingival crevicular fluid.
The main findings indicate that the clinical measurements associated with periodontal status were significantly higher in the older group compared to the adult group. Although there were no statistically significant differences in cytokine levels between the two groups, a strong correlation was observed between the clinical measurements and cytokine levels in the gingival crevicular fluid. Additionally, the older group exhibited an increasing tendency of both pro-inflammatory and anti-inflammatory cytokines compared to the adult group.
The strengths of this paper lie in its clinical approach, analyzing both the periodontal status and cytokine levels in an aging model. The inclusion of both adults and older individuals with periodontitis provides valuable insights into the impact of aging on periodontal health. Moreover, the strong correlation found between clinical measurements and cytokine levels reinforces the importance of these factors in understanding the inflammatory processes associated with periodontitis in older individuals.
General concept comments
A methodical opportunity would be that microbiome analyses were performed of the GCF to match the cytokine data with the clinical and microbiome data. This would be an interesting approach for future studies.
Specific comments
Introduction:
I consider using the word bacterial biofilms instead of plaque since it is more scientific. In sentence one I recommend to replace “plaque” by dysbiotic biofilms (line 27). Sentence 2 ff subsequently.
Please add one or two sentences about the process of “inflammaging” to the introduction, even though it is mentioned in the discussion. It is important to know at the beginning, that persistent activation of the innate immune system contributes to the development of chronic immune conditions commonly observed in older individuals. Notably, inflammaging not only diminishes the ability to mount an effective adaptive immune response but also leads to an exaggerated innate immune response, often referred to as a "cytokine storm."
Results:
Figure 1 and Figure 2 – the text in the figures seems blurred, please upload figures with the appropriate resolution
Literature:
Literature: Please check all cited articles. Some are in capital letters, some not. Ideally it should be all in small letters as mentioned in the authors instructions:
Journal Articles:
1. Author 1, A.B.; Author 2, C.D. Title of the article. Abbreviated Journal Name Year, Volume, page range.
The quality of English language is appropriate, nevertheless minor editing could further improve the overall quality of the paper.
Author Response
Dear reviewer
Thank you for your comments and suggestions on our manuscript. Below we provide clarifications and corrections in our response.
Point 1: General concept comments: A methodical opportunity would be that microbiome analyses were performed of the GCF to match the cytokine data with the clinical and microbiome data. This would be an interesting approach for future studies.
Response 1: Thank you very much for your insight. We have added this suggestion in the discussion part (page 8 line 338-340).
Point 2: Specific comments in Introduction:
- I consider using the word bacterial biofilms instead of plaque since it is more scientific. In sentence one I recommend to replace “plaque” by dysbiotic biofilms (line 27). Sentence 2 ff subsequently.
- Please add one or two sentences about the process of “inflammaging” to the introduction, even though it is mentioned in the discussion. It is important to know at the beginning, that persistent activation of the innate immune system contributes to the development of chronic immune conditions commonly observed in older individuals. Notably, inflammaging not only diminishes the ability to mount an effective adaptive immune response but also leads to an exaggerated innate immune response, often referred to as a "cytokine storm."
Response 2: We have replaced the word “plaque” by dysbiotic biofilms (from page 1 line 30, subsequently). We have added more explanation about the process of “inflammaging” in the introduction part according to your suggestion (page 2 line 70-77).
Point 3: Specific comment in Results: Figure 1 and Figure 2 – the text in the figures seems blurred, please upload figures with the appropriate resolution.
Response 3: We have uploaded the figures with appropriate resolution to the results.
Point 4: Specific comment in Literature: Please check all cited articles. Some are in capital letters, some not. Ideally it should be all in small letters as mentioned in the authors instructions:
Journal Articles:
- Author 1, A.B.; Author 2, C.D. Title of the article. Abbreviated Journal NameYear, Volume, page range.
Response 4: We have edited the cited articles as mentioned in the authors instructions.
Reviewer 2 Report
Dear authors:
This is a potentially well-designed study into the impact of aging on inflammatory conditions and the status of periodontal tissue.
I have some comments on it.
1. The sample size of your manuscript is too low, and we cannot make any definitive adjudication about the impact of aging on inflammatory cytokines in the periodontium, please more discuss it in discussion part.
2. Please rewrite the introduction part with more newer and relevant articles.
3. Please discuss more about the impact of the sexuality status of participants on your study result (your sample size of this study is too low).
4. Please mention the exact scientific source of method of GCF sample collection.
5. You evaluated only three different types of cytokines in your study, and we cannot extend this result to many other inflammatory cytokines in periodontium.
6. You mention in conclusion " The results of this study provide new insights into cytokine dynamics in periodontitis." But with a simple search in the scientific data base, we find a lot of similar articles in this area and your study does not have any novelty in this field.
The English language writing requires moderate native editing.
Author Response
Dear reviewer
Thank you for your comments and suggestions on our manuscript. Below we provide clarifications and corrections in our response.
Point 1: The sample size of your manuscript is too low, and we cannot make any definitive adjudication about the impact of aging on inflammatory cytokines in the periodontium, please more discuss it in discussion part.
Response 1: We have added the calculation of the sample size in materials and methods part (page 2 line 95-99). This was due to the limitations of our study. The study was done during the COVID-19 pandemic and additional charge for sample screening (PCR test for COVID-19) was insufficient, so the sample size of our manuscript was low. We have mentioned about our study limitation (page 8 line 338-340) and have discussed more the impact of aging on inflammatory cytokines in the periodontium in discussion part (page 7-8 line 271-327).
Point 2: Please rewrite the introduction part with more newer and relevant articles.
Response 2: We have revised the introduction part with newer and more relevant studies accordingly.
Point 3: Please discuss more about the impact of the sexuality status of participants on your study result (your sample size of this study is too low).
Response 3: We have added the impact of gender to inflammaging based on previous studies (page 6 line 231-236). However, we did not assess the relationship between gender and aging in our study.
Point 4: Please mention the exact scientific source of method of GCF sample collection.
Response 4: We have referred the scientific sources of method of GCF sample collection in the material and methods part according to your suggestion (page 3 line 131-138).
Point 5: You evaluated only three different types of cytokines in your study, and we cannot extend this result to many other inflammatory cytokines in periodontium.
Response 5: We have revised the title from the term “Periodontal Inflammatory Cytokines” to specific investigated cytokine. The new title become “Changes in Interleukin-1beta, Tumor Necrosis Factor-alpha, and Interleukin-10 Cytokines in Older People with Periodontitis”
Point 6: You mention in conclusion "The results of this study provide new insights into cytokine dynamics in periodontitis." But with a simple search in the scientific data base, we find a lot of similar articles in this area and your study does not have any novelty in this field.
Response 6: We have revised the statement into “The result of this study demonstrate cytokine dynamics of IL-1β, TNF-α, and IL-10 in periodontitis especially in older people” in conclusion part (page 8 line 343-344).
Round 2
Reviewer 2 Report
Dear authors:
The revised version of the manuscript was significantly improved, and I have no comments on it.
It is suitable.